# An Enhanced Joint Hilbert Embedding-Based Metric to Support Mocap Data Classification with Preserved Interpretability

**DOI:** 10.3390/s21134443

**Published:** 2021-06-29

**Authors:** Cristian Kaori Valencia-Marin, Juan Diego Pulgarin-Giraldo, Luisa Fernanda Velasquez-Martinez, Andres Marino Alvarez-Meza, German Castellanos-Dominguez

**Affiliations:** 1Faculty of Engineering, Universidad Tecnológica de Pereira, Pereira 660003, Colombia; 2G-Bio Research Group, Automatic and Electronic Department, Universidad Autónoma de Occidente, Cali 760030, Colombia; jdpulgarin@uao.edu.co; 3Signal Processing and Recognition Group, Universidad Nacional de Colombia sede Manizales, Manizales 170001, Colombia; lfvelasquezm@unal.edu.co (L.F.V.-M.); amalvarezme@unal.edu.co (A.M.A.-M.); cgcastellanosd@unal.edu.co (G.C.-D.)

**Keywords:** Hilbert embedding, joint distribution, time series, classification, Mocap data

## Abstract

Motion capture (Mocap) data are widely used as time series to study human movement. Indeed, animation movies, video games, and biomechanical systems for rehabilitation are significant applications related to Mocap data. However, classifying multi-channel time series from Mocap requires coding the intrinsic dependencies (even nonlinear relationships) between human body joints. Furthermore, the same human action may have variations because the individual alters their movement and therefore the inter/intraclass variability. Here, we introduce an enhanced Hilbert embedding-based approach from a cross-covariance operator, termed EHECCO, to map the input Mocap time series to a tensor space built from both 3D skeletal joints and a principal component analysis-based projection. Obtained results demonstrate how EHECCO represents and discriminates joint probability distributions as kernel-based evaluation of input time series within a tensor reproducing kernel Hilbert space (RKHS). Our approach achieves competitive classification results for style/subject and action recognition tasks on well-known publicly available databases. Moreover, EHECCO favors the interpretation of relevant anthropometric variables correlated with players’ expertise and acted movement on a Tennis-Mocap database (also publicly available with this work). Thereby, our EHECCO-based framework provides a unified representation (through the tensor RKHS) of the Mocap time series to compute linear correlations between a coded metric from joint distributions and player properties, i.e., age, body measurements, and sport movement (action class).

## 1. Introduction

Time series classification is a real-world problem that frequently deals with vast quantities of numerical measurements acquired at regular time intervals, having applications in fields such as share markets, biomedicine, intelligent sensor networks, and dynamic objects, among others [1,2,3,4]. Thus, in the case of moving objects, a contour of a static object can be transformed into a time series representation to favor image-based object recognition tasks [5,6,7]. Moreover, when classifying time series, one of the essential tasks is recognizing human actions. Most applications focused on the recognition of human activities are based on the construction of 3D skeletons composed of the human body joints extracted from computer vision systems using traditional video cameras (Microsoft Kinect and similar devices) [8]. However, these systems suffer from optical phenomena that affect their precision, such as changes in lighting and occlusions [9]. Then, to improve human pose tracking, there is considerable interest in techniques that avoid using a video camera—for example, WiFi human sensing [10] and radio-frequency identification (RFID) tags [11]. On the other hand, there are alternative methodologies based on holographic interferometry [12,13] that are remarkably robust to deformations and allow the skeletons of subjects to be adequately represented.

Regarding the motion capture (Mocap)-based human action analysis, different applications involve the classification of Mocap datasets, such as animation movies and video games [14], biomechanics systems for rehabilitation [15], and translation of sign languages [16], among others. However, Mocap data pose some issues for classifying human activities from time series. First, there is a need to code the time series dependencies (relationships between Mocap joints) to highlight discriminative patterns [17]. Second, a performance of a particular activity may have variations, which can be the results of individuals’ alteration of expression, posture, motion, and perspective effects [18]. In addition, the same sequence can be executed in different ways (styles) by distinct subjects [19]. Third, the Mocap data trajectories, obtained from 3D skeletal representations, are coded on high-dimensional spaces holding non-stationary dynamics [20].

In the literature, two main approaches are used to deal with time series representation and classification tasks: model-based (MB) and distance-based (DB) methods [2]. MB allows coding the temporal dependencies between time series from a set of parameters associated with a given stochastic or deterministic model. Some of the relevant examples include the hidden Markov models (HMMs) [21], the adaptive filters (AFs) [22,23], the Gaussian processes (GPs) [24,25], and deep networks [26,27]. HMM represents the input data from a sequence of hidden states that encode temporal dependencies among samples; nevertheless, an appropriate choice of the model’s topology/architecture is required, e.g., the covariance matrix shape and the number of hidden states [28]. In the case of AFs, they allow recursive learning of the time series, giving prominence to the most relevant data samples [29]. However, the quantization size and the error tolerance must be tuned appropriately, which can be problematic for 3D skeletal-based samples [30]. Regarding the GP-based methods, a Bayesian representation of time series is carried out. Although GPs are considered nonparametric models, their training is often computationally expensive when calculating the posterior distribution [25]. Recently, deep learning methods have been used for Mocap data classification [26,27,31]. Even though the classification performance is reasonable, exhaustive training is required, the overfitting issue arises for small databases, and the provided algorithms often lack straightforward interpretability [32].

Now, DB approaches reside in the construction of a dissimilarity space from the input time series, which are later used to train a classifier, e.g., a K-nearest neighbors [33,34]. In general, the Euclidean distance (ED) is the most straightforward DB approach. Nonetheless, ED can only be applied to discriminate time series of the same length [35]. Therefore, the dynamic time warping (DTW) dissimilarity appears as an extension of the ED, also known as 2-norm-based distance, to compare series of different lengths [36]. The DTW is quite well known for discriminating time series as it can be seen as a generalization of the ED exclusively for this kind of data [37]. Nevertheless, DTW requires crucial hyperparameter (warping percentage) tuning, and l2-based approaches tend to fail when coding nonlinear patterns [36]. In turn, reproducing kernel Hilbert space (RKHS)-based approaches have been proposed to highlight nonlinear data relationships [38]. Furthermore, Hilbert embedding-based dissimilarities have been introduced in the literature as a generalization of traditional kernel methods, mapping the input data probability distribution as a vector/operator in RKHS. The latter favors the estimation of dissimilarity-based measures within high dimensional spaces [39]. Of note, the Lie group representation approach is commonly applied on skeletal action recognition tasks [40,41,42]. However, Lie group-based methods suffer from temporal misalignment, which tends to deteriorate the classification accuracy [31]. To solve this problem, the DTW is coupled with the Lie group; nonetheless, the computational time is increased, and a two-step algorithm typically performs worse than an end-to-end learning strategy [31].

In this paper, an enhanced Hilbert embedding-based framework is proposed as a DB approach to support Mocap data classification. In this sense, a novel metric is introduced to map joint probability distributions, from two different input spaces, in a tensor RKHS through the cross-covariance operator [43,44]. Our approach, termed enhanced Hilbert embedding from cross-covariance operator (EHECCO), allows comparing input data from sample-based kernel evaluations, circumventing the direct estimation of probability functions. The latter helps in the analysis of multi-view instances in pattern recognition tasks, i.e., classification from data fusion [45]. Then, we aim to code temporal information from sequentiality data to support further classification stages regarding human action recognition (HAR). The most significant contributions in this work can be summarized as follows: (i) a novel analytical expression for calculating an RKHS-based dissimilarity to discriminate between joint probability distributions; (ii) a representation strategy for the extraction and processing of skeletons from Mocap videos, which allows finding the most relevant and discriminating movement patterns; and (iii) a recognition framework of human activities and style based on EHECCO, which allows anthropometric analysis and proper interpretation of the results obtained. Indeed, our EHECCO-based framework for HAR facilitates the computation of linear correlations between the coded metric, player properties (age, body measurements, among others), and human action classes. Of note, EHECCO can deal with different time series lengths, preserving the most relevant frames (human poses) when comparing the Mocap time series. Our method is a crucial improvement compared with conventional human movement analysis approaches, which employ alienation angles, linear velocities, and angular velocities as factors to be evaluated [46]. The approach is tested on both public (for action and style recognition) and our own (for action recognition and anthropometric analysis) Mocap datasets. Results obtained are competitive in terms of the achieved classification accuracy with the benefit of Mocap data interpretability.

The remainder of this paper is organized as follows: Section 2 describes the mathematical background. Section 3 shows the experimental set-up. Section 4 presents the results and discussion. Finally, the conclusions appear in Section 5.

## 2. Methods

In this section, we provide the mathematical background concerning our Hilbert embedding-based metric. First, the well-known marginal embedding approach is briefly described. Then, we present our joint embedding proposal to build a metric in a tensor RKHS from joint distributions. Our approach seeks to exploit two main issues: (i) joint distribution-based modeling from two different input spaces, and (ii) non-linear sample mapping to code relevant data dependencies from joint distributions circumventing the direct estimation of probability functions. The latter would be helpful to deal with multi-channel time series, which is the basis of our experimental set-up concerning HAR from Mocap data.

### 2.1. Marginal Embedding-Based Metric in RKHS

Let $P_{X}$ be the space of all marginal probability distributions on
X. Moreover, let *X* be a random variable with distribution $P_{X}\in P_{X}$. A marginal embedding μH X∈H can be defined as [47]:
(1)$$\mu_{H}^{X}=E_{x}\left[\varphi \left(x\right)\right]=\int_{X}\varphi \left(x\right)dP_{X},$$
where x∈X is a given sample and
H is a reproducing kernel Hilbert space (RKHS) holding the nonlinear mapping φ:X→H. E[·] stands for the expectation operator. Furthermore, let *Z* be another random variable with distribution $P_{Z}\in P_{X}$ and marginal embedding $\mu_{H}^{Z}\in H.$ Then, a distance metric $d:P_{X}\times P_{X}\to R^{+}$ between probability distributions can be defined in H from the marginal embeddings $\mu_{H}^{X}$ and $\mu_{H}^{Z}$ as:
(2)$$d^{2}\left(P_{X},P_{Z}\right)={∥\mu_{H}^{X}-\mu_{H}^{Z}∥}_{H}^{2},$$
where ${∥\cdot ∥}_{H}$ stands for the norm operator in H. Founded on the kernel trick property $\kappa_{\varphi }\left(x,x^{\prime }\right)={\langle \varphi \left(x\right),\varphi \left(x^{\prime }\right)\rangle }_{H}$, being $\kappa_{\phi }:X\times X\to R$ a positive semi-definite characteristic kernel function [39], the metric in Equation (Equation 2) can be rewritten as [48]:
(3)$$d^{2}\left(P_{X},P_{Z}\right)=E_{x,x^{\prime }}\left[\kappa_{\varphi }\left(x,x^{\prime }\right)\right]+E_{z,z^{\prime }}\left[\kappa_{\varphi }\left(z,z^{\prime }\right)\right]-2E_{x,z}\left[\kappa_{\varphi }\left(x,z\right)\right],$$
with $x,x^{\prime }\in X$ and $z,z^{\prime }\in Z.$

The expression in Equation (Equation 3) is an analytical metric function in RKHS for probability distributions [49]. In fact, the well-known maximum mean discrepancy (MMD) distance arises from Equation (Equation 3) to extend traditional kernel methods for estimating probability functions [22,50]. Namely, let ${\left\{x_{n}\in X\right\}}_{n=1}^{N}$ and ${\left\{y_{m}\in X\right\}}_{m=1}^{M}$ be a pair of sets holding *N* and *M* samples, respectively. Moreover, let us assume that the probability distributions $P_{X}$ and $P_{Z}$ admit density functions p(x) and p(y). Then, after fixing the empiric-based estimators $\hat{p}\left(x\right)=\frac{1}{N}\sum_{n=1}^{N}\delta \left(x-x_{n}\right)$ and $\hat{p}\left(y\right)=\frac{1}{M}\sum_{m=1}^{M}\delta \left(y-y_{m}\right)$, δ(·)∈{0,1} stands for the delta function, and using a Gaussian characteristic kernel $\kappa_{\sigma }\left(x_{n},y_{m}\right)=\exp(-∥x_{n}-y_{m}{∥}_{2}^{2}/2\sigma^{2})$, $\sigma \in R^{+}$ is a similarity bandwidth, the MMD estimator is given by [51]:
(4)$${\hat{d}}_{\mathrm{MMD}}^{2}\left(P_{X},P_{Z}\right)=\frac{1}{N^{2}}1_{N}^{⊤}K^{x,x}1_{N}+\frac{1}{M^{2}}1_{M}^{⊤}K^{y,y}1_{M}-\frac{2}{NM}1_{N}^{⊤}K^{x,y}1_{M},$$
where
$K^{x,x}\in R^{N\times N},$
$K^{y,y}\in R^{M\times M},$ and
$K^{x,y}\in R^{N\times M}$ are kernel matrices computed from
$\kappa_{\sqrt{2}\sigma }\left(\cdot ,\cdot \right).$
$1_{N}$ and
$1_{M}$ are all one column vectors of size *N* and
M, respectively.

### 2.2. Enhanced Hilbert Embedding from Cross-Covariance Operator (EHECCO)

Though MMD in Equation (Equation 4) allows comparing samples without any assumption over probability distributions, it only codes the marginal information when performing the distance-based representation. Therefore, dealing with complex data relationships—for example, Mocap time series classification for HAR—will benefit from representing the instances on different RHKS to code contrasting properties of the samples. Then, a joint distribution-based metric can be developed.

Let us consider another pair of random variables
Y,L∈Y with distributions
$P_{Y},P_{L}\in P_{Y}$, where
$P_{Y}$ is the space of all marginal distributions on
Y; further, let
y∈Y and
l∈L be samples from the aforementioned random variables. Our enhanced Hilbert embedding from cross-covariance operator (EHECCO) allows computing a metric between the joint distributions
$P_{X,Y},P_{Z,L}\in P_{X,Y},$ where
$P_{X,Y}$ is the space of all joint probability distributions defined on the Cartesian product
X×Y. Following the metric in Equation (Equation 2), the RKHS-based distance
$d_{J}:\left(P_{X,Y}\times P_{X,Y}\right)\times \left(P_{X,Y}\times P_{X,Y}\right)\to R^{+}$ between joint probability distributions yields:
(5)$${d_{J}}^{2}\left(P_{X,Y},P_{Z,L}\right)={∥\mu_{H⊗G}^{X,Y}-\mu_{H⊗G}^{Z,L}∥}_{H⊗G}^{2},$$
where the Hilbert embeddings
$\mu_{H⊗G}^{X,Y},\mu_{H⊗G}^{Z,L}\in H⊗G$, being
H⊗G a tensor space, can be defined as the following cross-covariance operators [48]:
(6)$$\begin{array}{c} \mu_{H⊗G}^{X,Y}=E_{X,Y}\left[\varphi \left(x\right)⊗\phi \left(y\right)\right], \end{array}$$
(7)$$\begin{array}{c} \mu_{H⊗G}^{Z,L}=E_{Z,L}\left[\varphi \left(z\right)⊗\phi \left(l\right)\right], \end{array}$$
where
φ(x),φ(z)∈H,ϕ(y),ϕ(l)∈G are nonlinear mappings to the RKHS
H and
G, following the positive semi-definite characteristic kernels:
$\kappa_{\varphi }\left(x,x^{\prime }\right)=\;{\langle \varphi \left(x\right),\varphi \left(x^{\prime }\right)\rangle }_{H}$,
$\forall x,x^{\prime }\in X$ and
$\kappa_{\phi }\left(y,y^{\prime }\right)={\langle \phi \left(y\right),\phi \left(y^{\prime }\right)\rangle }_{G},\forall y,y^{\prime }\in Y$ [49]. The latter is accomplished too for samples of the random variables *Z* and *L*, respectively.

Furthermore, let us assume that
$P_{X,Y}$ and
$P_{Z,L}$ admit density functions
p(x,y) and
p(z,l), respectively; then,
$dP_{X,Y}=p\left(x,y\right)dxy$ and
$dP_{Z,L}=p\left(z,l\right)dzl$. We can rewrite Equation (Equation 5) as follows [52]:
(8)$$\begin{array}{cc} {d_{J}}^{2}\left(P_{X,Y},P_{Z,L}\right)= & \int_{X\times Y}\int_{X\times Y}\kappa_{\varphi }\left(x,x^{\prime }\right)\kappa_{\phi }\left(y,y^{\prime }\right)p\left(x,y\right)p\left(x^{\prime },y^{\prime }\right)dxydx^{\prime }y^{\prime }\\ \; & +\int_{X\times Y}\int_{X\times Y}\kappa_{\varphi }\left(z,z^{\prime }\right)\kappa_{\phi }\left(l,l^{\prime }\right)p\left(z,l\right)p\left(z^{\prime },l^{\prime }\right)dzldz^{\prime }l^{\prime }\\ \; & -2\int_{X\times Y}\int_{X\times Y}\kappa_{\varphi }\left(x,z\right)\kappa_{\phi }\left(y,l\right)p\left(x,y\right)p\left(z,l\right)dxydzl. \end{array}$$

Of note, the metric presented in Equations (Equation 5) and (Equation 8) (see Figure 1 for a schematic illustration) favors the extraction of relevant patterns from joint distributions as vector-based mappings in RKHS. Indeed, Hilbert embedding-based feature representations allow mapping marginal, conditional, and joint distributions into feature spaces using kernels, comparing and manipulating these distributions via feature space operations [44]. Our proposal is a direct extension of the conventional marginal embedding approach presented in Equation (Equation 2) towards a metric between joint distribution (see Theorem 1 in [48]). Moreover, it is well known in the machine learning literature that kernel-based methods favor highlighting nonlinear dependencies from input samples by mapping them to high-dimensional, possibly infinite, Hilbert space, revealing discriminative data patterns [53].

For concrete testing, let
${\left\{x_{n}\in R^{V},y_{n}\in R^{Q}\right\}}_{n=1}^{N}$ and
${\left\{z_{m}\in R^{V},l_{m}\in R^{Q}\right\}}_{m=1}^{M}$ be a pair of input sets (time series coded into two different spaces), and our matrix-based estimator in Equation (Equation 8) yields:
(9)$${{\hat{d}}_{J}}^{2}\left(P_{X,Y},P_{Z,L}\right)\;=\;\alpha_{x,y}^{⊤}\left(K_{\varphi }^{x,x}\circ K_{\phi }^{y,y}\right)\alpha_{x,y}\;+\;\alpha_{z,l}^{⊤}\left(K_{\varphi }^{z,z}\circ K_{\phi }^{l,l}\right)\alpha_{z,l}\;-2\alpha_{x,y}^{⊤}\left(K_{\varphi }^{x,z}\circ K_{\phi }^{y,l}\right)\alpha_{z,l},$$
where the kernel matrices
$K_{\varphi }^{x,x},K_{\phi }^{y,y}\in^{N\times N},$
$K_{\varphi }^{z,z},K_{\phi }^{l,l}\in^{M\times M},$ and
$K_{\phi }^{x,z},K_{\varphi }^{y,l}\in^{N\times M}$ are computed based on the kernel functions
$\kappa_{\varphi }\left(\cdot ,\cdot \right)$ and
$\kappa_{\phi }\left(\cdot ,\cdot \right)$. The operator ∘ stands for the Hadamard product. Moreover, the probability column vectors $\alpha_{x,y}\in {\left[0,1\right]}^{N}$ and
$\alpha_{z,l}\in {\left[0,1\right]}^{M}$ hold the joint probability estimators
$\hat{p}\left(x_{n},y_{n}\right)$ and
$\hat{p}\left(z_{m},l_{m}\right),$ respectively.

It is worth mentioning that our EHECCO estimator in Equation (Equation 9) provides a data-driven metric in the tensor space
H⊗G to compare the joint distributions
$P_{X,Y}$ and
$P_{Z,L}$ as kernel-based operations of input vectors. Remarkably, it can benefit further classification stages by extracting discriminative features from high-dimensional feature spaces through our kernel-based approach.

In short, our EHECCO-based metric seeks to exploit two main issues: (i) joint distribution-based time series modeling from two different input spaces, and (ii) non-linear data mapping to code relevant sample dependencies from joint distributions, circumventing the direct estimation of probability functions. Regarding the classification of multi-channel time series, i.e., HAR based on Mocap records, spatio-temporal relationships can be highlighted from the joint space (tensor RKHS), favoring data discrimination. Moreover, as our EHECCO-based metric can deal with different time series lengths, the most relevant frames (human poses) can be preserved when comparing time series. The latter is a crucial improvement compared with conventional human movement analysis approaches, which employ alienation angles, linear velocities, and angular velocities as factors to be evaluated [46].

## 3. Experimental Setup

Our EHECCO metric in Equation (Equation 9) is used to construct a HAR framework from Mocap videos. Thereby, we aim to demonstrate the discriminative capability and interpretability benefits of our joint distribution-based embedding approach to deal with multi-channel time series related to human movement. Then, the experimental design of our EHECCO-based framework can be summarized in the following stages:
–3D joint normalization. A 3D joint representation is extracted from each Mocap record followed by a hip-based normalization [27].–Codebook generation. A codebook of Mocap frames is built to gather the most representative movement poses. Then, a set of Nc clusters is computed using the well-known spectral clustering algorithm [54], from a vector-based concatenation of the 3D joints. The radial basis function is used as similarity, fixing the bandwidth as the median of the input Euclidean distances.–Joint and latent space-based representations. To code relevant patterns from provided codebooks, both the input joints and their latent space are considered to build a Mocap video input set:
${\left\{x_{n}\in R^{V},y_{n}\in R^{Q}\right\}}_{n=1}^{N_{c}}$. Here, the well-known principal component analysis (PCA) algorithm is employed to compute a latent space coding the most relevant orthonormal basis concerning the preserved input channels’ variability [55]. In fact, for concrete testing, three principal components are considered (Q=3). According to our experiments, three components preserve at least
75% of the input data variability. Note that the *V* value equals the number of Mocap joints times three (3D skeleton).–EHECCO-based dissimilarity representation and classification. Given a a pair of Mocap video sets:
${\left\{x_{n}\in R^{V},y_{n}\in R^{Q}\right\}}_{n=1}^{N_{c}}$,
${\left\{z_{m}\in R^{V},l_{m}\in R^{Q}\right\}}_{m=1}^{N_{c}}$, our EHECCO-based distance measure in Equation (Equation 9) is computed. In turn, a dissimilarity matrix
$D\in R^{\Lambda \times \Lambda }$ is calculated as EHECCO-based pairwise Mocap video comparisons (
Λ stands for the number of processed Mocap videos). For the tested databases, the probability vectors are fixed as
$\alpha_{x,y},\alpha_{z,l}\;∼U\left[0,N_{c}\right]$, being
$U[0,N_{c}]$ the uniform distribution. Since the Gaussian kernel is preferred in pattern classification because of its universal approximating ability and mathematical tractability [56], $\kappa_{\varphi }\left(\cdot ,\cdot \right)$ and
$\kappa_{\phi }\left(\cdot ,\cdot \right)$ are fixed as Gaussians. Each kernel bandwidth is searched within the range
$\left\{0.5\sigma_{0},\sigma_{0},2\sigma_{0},5\sigma_{0},10\sigma_{0}\right\}$ concerning the final classification performance.
$\sigma_{0}\in R^{+}$ equals the median of input Euclidean distances in accordance with each studied space
X (input Mocap joints) or
Y (PCA-based latent projection). Finally, a support vector machine (SVM) classifier is trained on the EHECCO’s distance matrix. A radial basis function (nonlinear mapping) is set for the SVM, and the penalty and precision hyper-parameters are settled from the grids
{1,10,100,1000,10,000} and
{0.01,0.1,1,100,1000}, respectively, concerning the classification performance. In addition, 2D data projection is also provided from the EHECOO metric for visualization purposes.

Figure 2 also summarizes the provided EHECCO-based flowchart for Mocap data classification.

### 3.1. Mocap Databases

For concrete testing, the following databases are tested for human action classification and analysis from the Mocap data:HDM05 for style/subject recognition (http://resources.mpi-inf.mpg.de/HDM05/, accessed on 5 October 2020). This database includes 325 records (from 65 actions) performed by five different subjects. The dataset includes several recorded actions using a Vicon mocap system, where 31 reflective markers are placed on the subject’s bodies [57]. Then, multi-channel time series of BVH files at 120 frames per second is provided. Following the framework proposed by the authors in [27], we built a scheme for style classification (subject recognition). We relate the classes to each of the five subjects who perform the actions as follows: subject 1 (s1) and similarly for the other subjects.CMU subset for action recognition (http://mocap.cs.cmu.edu/info.php, accessed on 5 October 2020). Mocap data are obtained from the Carnegie Mellon Graphics Laboratory, holding 12 Vicon infrared MX-40 cameras at 120 Hz with images of four-megapixel resolution. The cameras are placed around a rectangular area, of approximately 3 m × 8 m, in the center of the room. In particular, multi-channel time series as BVH files with 38 markers are provided. In the same way, as in [26], an action recognition task is carried out from a subset of 150 clips of 15 different motion classes (performed by several subjects): *walking (wal), running (run), sitting (sit), jumping (jum), weight-carrying (wei), climbing (cli), swinging (swn), placing a ball (plb), placing tee (plt), kicking (kic), soccer and basketball playing (soc), boxing (box), swimming (swm), salsa (sal)*, and *Indian Bollywood dancing (InB).*Tennis-Mocap for action recognition and anthropometric analysis (https://drive.google.com/file/d/1-3HAUP4vIBBMz21f7RRgA4b89uNrLxvr/view?usp=sharing, accessed on 5 October 2020). The data are collected from 17 players of the Caldas-Colombia tennis league. The employed motion capture protocol includes the placement of 34 markers for collecting information on body joints. Optitrack Flex V100 (100 Hz) infrared videography is collected from six cameras to acquire sagittal, frontal, and lateral planes. All subjects are encouraged to hit the ball with the same velocity and action as in a tennis match. Moreover, the players are instructed to hit one series continuously by 30 s of each indicated stroke: *serve (Ser), forehand (For), backhand (Bac), volley (Vol), backhand volley (BaV)*, and *smash (Sma)*. In addition, the Tennis database includes the anthropomorphic players’ measurements depicted in Table 1.

### 3.2. Method Comparison, Quality Assessment, and Implementation Details

To evaluate the performance of our EHECCO-based framework to classify Mocap data, we compare the results on the public databases (HDM05 and CMU subset) obtained in HAR with relevant state-of-the-art approaches:

Method comparison for HDM05 dataset (style/subject recognition). We compare our own method with the following methods: symmetric positive definite network (SPDNet) [40], special Euclidean group (SE) [41], special orthogonal group (SO) [42], Lie groups on deep neural networks (LieNet) [31], and works based on 3D sequence to RGB image transformation (Seq2im) [27].

Method comparison for CMU subset (action recognition). We compare our results with the following approaches: motion template combined with a DTW-based classifier (MT+DTW) [58], self-similarity matrix with DTW distance (SSM+DTW) [18], efficient motion retrieval (EMR) [59], and motion words with convolutional neural networks (MW+CNN) [26].

Afterward, regarding the Tennis-Mocap database (own database), we carried out action recognition tasks along with anthropometric analysis using the extracted EHECCO-based patterns together with the measurements presented in Table 1.

As a quality assessment, we use a 10-fold cross-validation strategy based on the well-known average accuracy and confusion matrix performance measures [54]. As an illustrative example, the accuracy for a binary classification case is defined as $A_{cc}=\left(T_{p}+T_{n}\right)/N$, where
$T_{p}$ and
$T_{n}$ are the true positive and true negative classifier’s predictions, respectively, being *N* the number of studied samples. Similarly, the confusion matrix for a binary classification task includes an array holding the values of
$T_{p}$ and
$T_{n}$ in the main diagonal and the false positive (
$F_{p}$) and false negative (
$F_{n}$) predictions on the upper and lower triangular matrix positions.

All our experiments are implemented in Python using the sklearn toolbox for the training and validation of the models and the PyMO library (https://github.com/omimo/PyMO, accessed on 5 October 2020) for the management and representation of Mocap data. The most relevant codes of this paper can be found in a publicly available repository (https://github.com/Ckvalencia/hello-world/blob/master/SHECCO_CMU_sub.ipynb, accessed on 12 April 2021).

## 4. Results and Discussion

This section describes the classification results obtained by EHECCO-based distance for the Mocap datasets specified in Section 3.1.

### 4.1. HDM05 and CMU Results: Mocap Classification Benchmark

Figure 3 presents an example of relevant skeletons (codebook generation) for a given Mocap video selected from HDM05 and CMU datasets. For illustration purposes, two classes are investigated: throwing high with the right hand while standing and boxing, for which the 2D PCA projection is dotted with colored points, while the recorded frames are pictured with black points. Note that the frames chosen by the clustering algorithm are distributed so that they cover the entire space. As seen, the algorithm manages to capture the most relevant information about the movement without significant loss of information. Furthermore, the *boxing* record results show how both the codebook generation and the PCA-based projection preserve the cyclic action behavior, e.g., the subject acting several times.

For each database, Figure 4 presents the confusion matrix along with the 2D low-dimensional scatter plot performed by the EHECCO distance matrix
D using the t-distributed stochastic neighbor embedding (t-SNE) algorithm [60]. The scatter plot visually interprets the EHECCO patterns, preserving the spatial relationships in the higher tensor space (nearest-neighbors) [61]. As a result, our EHECCO approach achieves a competitive discrimination performance concerning both subject/style and action recognition tasks, reaching an average accuracy of
88.8 and 90 percentage in HDM05 and CMU subsets, respectively. The scatters also evidence the EHECCO’s ability to reveal both local and global data patterns. Of note, some classes hold nonstationary behavior, due to groups overlapping, i.e., see the confusion matrices and the 2D projections for subject two vs. subject five in HDM05: *sal* vs. *cli*, *soc* vs. *sit*, and *plb* vs. *kic* actions for the CMU subset. The behavior of this latter paired comparison is expected because of the Mocap data variations [19]. Overall, the combination of EHECCO with SVM can deal with the intra/interclass variability.

One more aspect to highlight is comparing the performance EHECCO classification performance with several state-of-the-art results recently reported. Thus, Table 2 shows the accuracy results for the *HDM05* dataset, including the following methods: symmetric positive definite network (SPDNet) [40], special Euclidean group (SE) [41], special orthogonal group (SO) [42], Lie groups on deep neural networks (LieNet) [31], and sequence to RGB image (Seq2im) [27]. The latter employs 3D sequence to RGB image transformation combined with conventional classifiers such as SVM, K-nearest neighbors (KNN), random forest (RF), and convolutional neural networks (CNN). As seen, the EHECCO+SVM combination overcomes the state-of-the-art techniques compared, including those based on deep learning such as Seq2im+CNN. Nevertheless, deep learning approaches often require exhaustive fine-tuning, whereas our EHECCO-based metric provides a data-driven technique as input vector evaluations for nonlinear pattern extraction in RHKS.

Furthermore, Table 3 presents the comparison results for the CMU subset, which includes the motion template (MT), self-similarity matrix (SSM), and efficient motion retrieval (EMR) methods [18,58,59], relying on dissimilarity matrices obtained from Mocap data feature extraction techniques and the DTW distance. Although they managed to obtain promising results, their achieved performance is not competitive enough concerning more recent methods. Motion word-(MW)-based methodology [26] yields competitive accuracy. In fact, MW incorporates a deep learning scheme to favor the time series representation. Our EHECOO outperforms most of the compared works, and it is rather similar regarding the achieved accuracy compared to the work proposed in [26]. Hence, EHECOO allows encoding nonlinear Mocap data similarities from both the 3D skeleton and PCA-based latent space through a joint distribution comparison perspective. Thereby, the EHECCO+SVM pipeline supports both the style and action recognition performance with the benefit of providing the metric interpretability of the extracted representation.

### 4.2. Tennis-Mocap Results: Classification and Anthropomorphic Analysis

Figure 5 depicts the codebook generation (relevant poses) for some videos of the Tennis-Mocap database. Usually, the alienation angles, linear velocities, and angular velocities are factors to be evaluated in the training of a professional tennis player [46,62]. Nevertheless, the analysis of the action execution is costly and involves kinetic analysis with additional instrumentation [63]. Our method shows a valuable tool based only on kinematic information provided by optical sensors. Indeed, our EHECCO-based approach allows encoding the relevant poses characterizing from the time series (tennis action) without any manual frame segmentation or preprocessing. As seen, the provided codebook encodes the most relevant information in the first execution of each record and some significant variations in the posterior executions of the action.

Regarding the classification results, as can be seen at the top of Figure 6, accuracies over 80% are attained. The lowest performance must be analyzed in conjunction with the action, where the upper limb’s position in the most relevant poses makes these classes closer. Nevertheless, each record classified contains 12 to 16 continuous stroke executions without segmentation, so the confused actions depend on the execution speed after 30 s. The latter can also be corroborated by the 2D t-SNE data projection, where both the action and the players’ expertise are presented. As seen, intra and interclass variability are revealed, corroborating the EHECCO’s ability to highlight nonlinear patterns related to the player’s performance (style/expertise) and the action behavior. However, movements such as smash, serve, and forehand involve a significant arm span in execution, being difficult to separate. Moreover, they involve major upper-body power/strength as referred in [64]. Though the “arm span” measure is used in anthropometric tennis studies, it has no statistical significance in the early stages when classifying competitive and non-competitive players [65].

Lastly, the bottom of Figure 6 displays the Pearson’s correlation-based analysis (absolute value) to compute the linear dependencies between the mean 1D t-SNE-based projection of the players’ samples (from EHECOO metric) and the Tennis-Mocap dataset anthropometric measurements (see Table 1). In particular, the correlation analysis is carried out concerning the six movements performed by the players to find the incidence of each physical variable in the execution of the studied actions.

As seen, fat fold variables are highly correlated with each other, similarly to the perimeter variables. Moreover, the tennis actions share substantial correlations with the players’ perimeter measurements (blue), specifically with the forehand, backhand, and volley classes. Notably, EHECOO-based interpretability follows the fact that anthropometric characteristics related to the size of the limbs and other parts of the body have a more significant influence on players’ performance than features related to age, weight, height, and strength [66].

## 5. Concluding Remarks

We introduced a new enhanced Hilbert embedding-based framework from a cross-covariance operator, termed EHECCO, to represent and discriminate joint probability distributions in RKHS. Our approach favors the extraction of relevant nonlinear dependencies from input vectors to support the time series classification. In this sense, an EHECCO-based framework is tested to support Mocap data classification concerning style/subject and action recognition as well as anthropometric analysis. The introduced framework includes a codebook generation and a PCA-based latent space extraction for coding the most relevant frames and patterns from the Mocap series. Then, our EHECCO-based metric is computed to feed an SVM classifier. Provided experiments include the well-known public databases *HDM05* and *CMU subset* and our own dataset, *Tennis-Mocap* (also publicly available). As shown, EHECCO obtains competitive classification performances for both style and action recognition, outperforming state-of-the-art approaches. Moreover, EHECOO codes the intra and interclass variability and favors the interpretation of relevant anthropometric variables correlated with subject expertise and performed actions.

As future work, the authors plan to include other anthropometric and sports measurements to enhance the proposed framework, i.e., the arm span will be more sensitive in elite players’ classification [65]. Moreover, EHECCO-based HAR applications from conventional video camaras [8], WiFi human sensing [10], and RFID [11] data will be carried out. Further, we plan to test the EHECCO metric on other types of time series, i.e., brain activity data [67]. Additionally, more elaborate classifiers and deep learning schemes can benefit from our EHECCO metric [68]. Finally, an extension of the EHECCO distance for the joint distribution of multiple spaces, not only two, is a research line of interest.

## Figures and Tables

**Figure 1 sensors-21-04443-f001:**
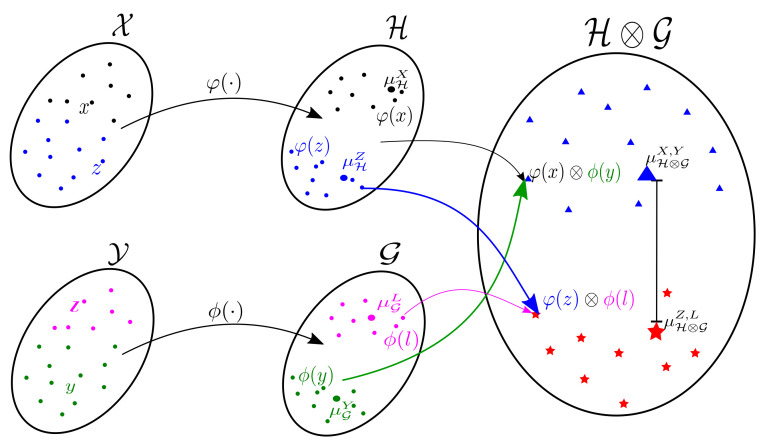
Schematic illustration of our EHECCO-based metric. Input spaces X and
Y are mapped to RKHSs
H and
G, respectively. Then, the tensor space
H⊗G is built using a cross-covariance operator strategy.

**Figure 2 sensors-21-04443-f002:**
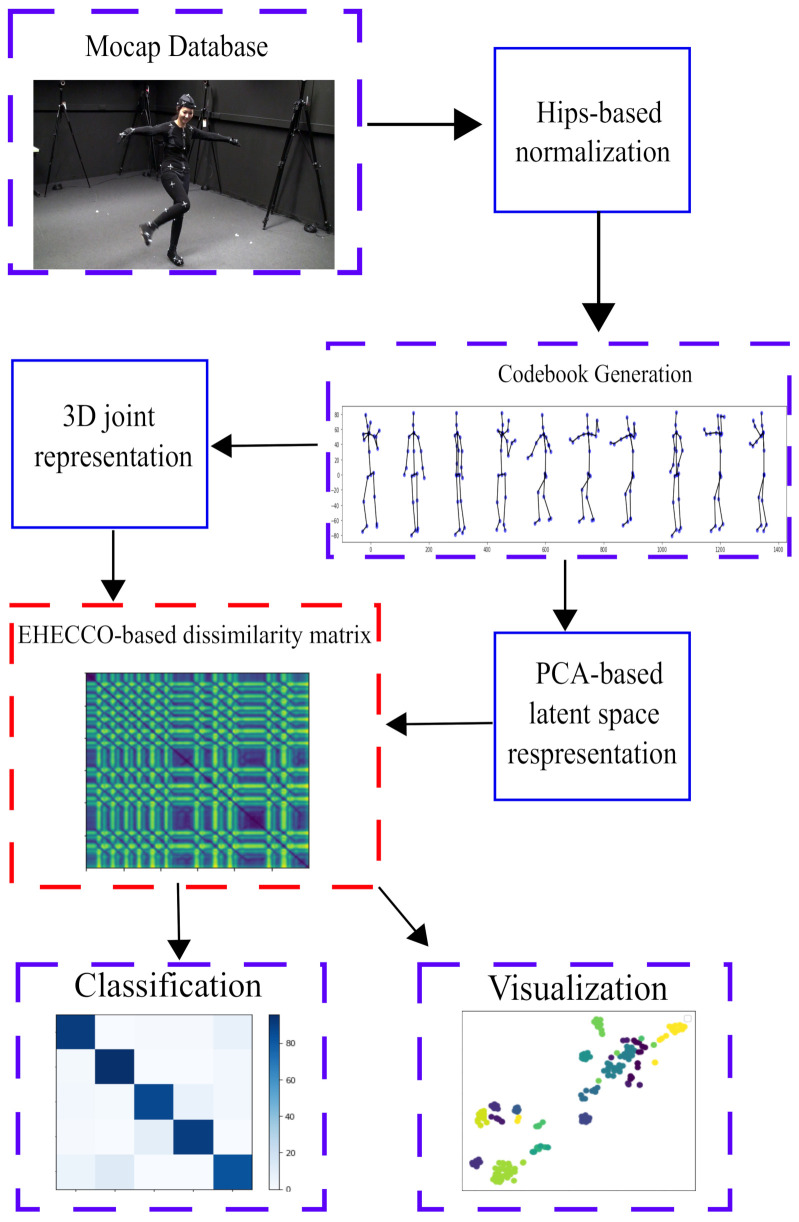
EHECCO-based Mocap data classification framework. Hip joint normalization and spectral clustering-based codebook generation are carried out to extract relevant skeletal poses. Then, 3D joint representation (X) and PCA-based latent projection (
Y) are used to support the EHECCO metric from joint probability. Lastly, an SVM classifier is trained from the EHECCO distance that also supports 2D data visualization.

**Figure 3 sensors-21-04443-f003:**
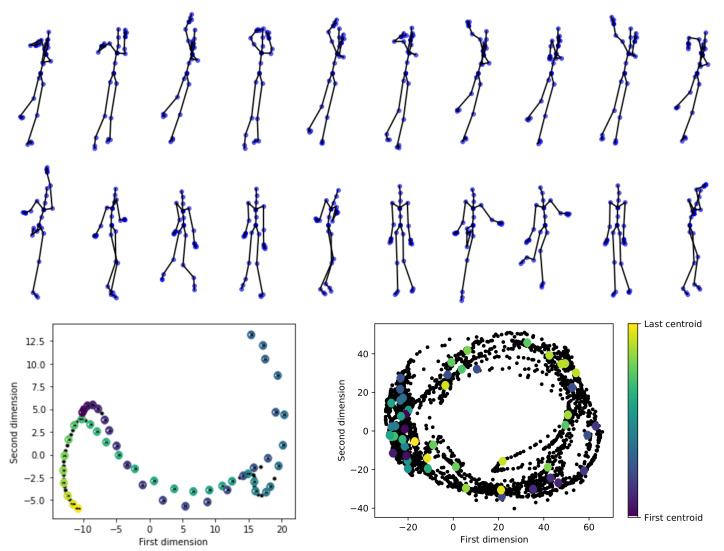
Illustrative results for codebook generation and latent space-based representation (HDM05 and CMU subset datasets). Top: Codebook generation for a Mocap video of the *throwing high with the right hand while standing* class (HDM05). Middle: Codebook generation for a Mocap record of *boxing* class (CMU subset). Bottom left: PCA-based latent space for HDM05 video. Bottom right: PCA-based latent space for CMU subset video. The first two components are shown for visualization purposes. Black markers represent the original input Mocap frames (time series). Color markers represent the chosen frames (codebook).

**Figure 4 sensors-21-04443-f004:**
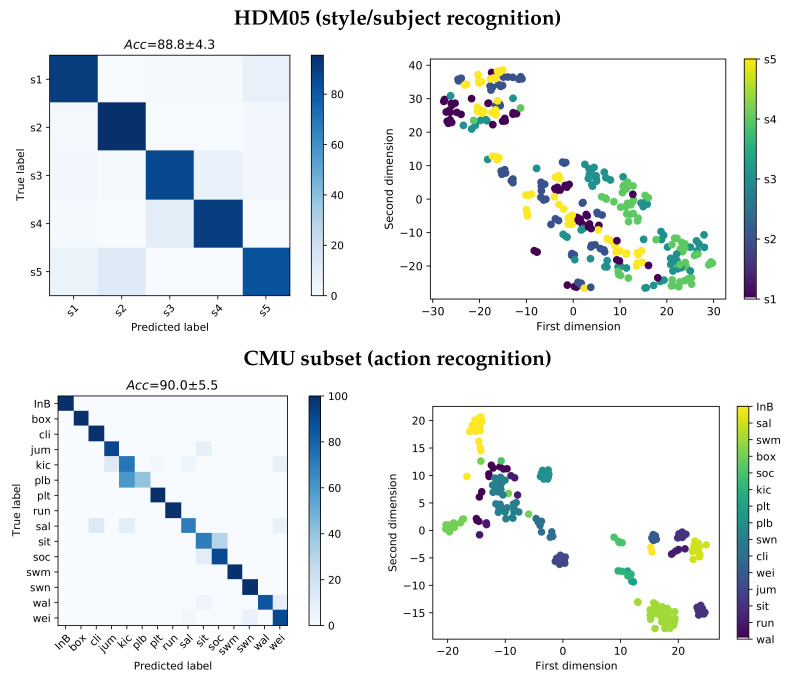
EHECCO-based classification results for HDM05 and CMU subset databases. **Top left**: HDM05’s confusion matrix (style/subject recognition). **Top right**: HDM05 t-SNE-based 2D projection from EHECCO distance. **Bottom left**: CMU subset’s confusion matrix (action recognition). **Bottom right**: CMU subset t-SNE-based 2D projection from EHECCO distance.

**Figure 5 sensors-21-04443-f005:**
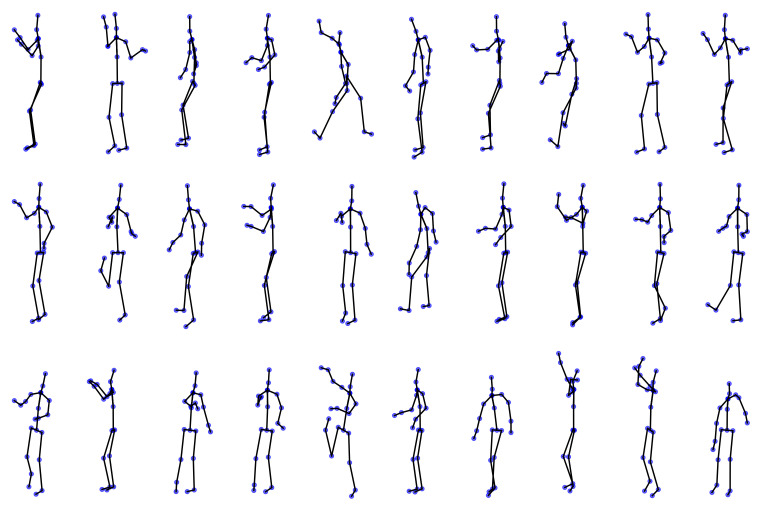
Illustrative results for codebook generation (Tennis-Mocap dataset). Top: *forehand*; Middle: *volley*; Bottom: *Smash*.

**Figure 6 sensors-21-04443-f006:**
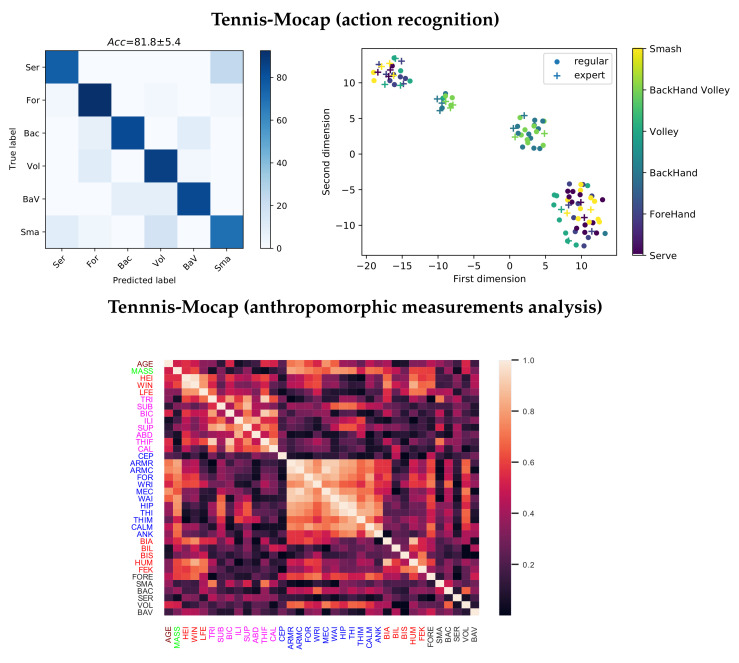
EHECCO-based classification and anthropomorphic measurement results for Tennis-Mocap database. **Top left**: confusion matrix (action recognition). **Top right**: t-SNE-based 2D projection from EHECCO distance. **Bottom left**: Absolute value of the Pearson’s correlation coefficient between the EHECCO first t-SNE-based mean projection of each player’s videos and his/her anthropomorphic measurements. The most relevant correlations are shown.

**Table 1 sensors-21-04443-t001:** Tennis dataset’s anthropomorphic measurements. The color represents the measurement group: age (brown), weight (light green), length (red), perimeters (blue), fat fold (pink), and tennis move (black).

Age	Thigh cm (THI)	Height cm (HEI)	Medial calf mm (CAL)
Mass	Calf maximum cm (CALM)	Foot length cm (LFE)	Biceps mm (BIC)
Cephalic cm (CEP)	Relaxed arm cm (ARMR)	Biliocrestal cm (BIL)	Front thigh mm (THIF)
Minimum ankle cm (ANK)	Mesosternal chest cm (MEC)	Humerus cm (HUM)	Forehand (FORE)
Hip max cm (HIP)	Forearm cm (FOR)	Supraspinal mm (SUP)	Smash (SMA)
Contracted arm 90 cm (ARMC)	Bistyloid cm (BIS)	Subscapular mm (SUB)	Backhand (BAC)
Waist cm (WAI)	Biacromial cm (BIA)	Iliac crest mm (ILI)	Serve (SER)
Middle thigh cm (THIM)	Femur knee cm (FEK)	Triceps mm (TRI)	Volley (VOL)
Wrist cm (WRI)	Wingspan cm (WIN)	Abdominal mm (ABD)	Backhand Volley (BAV)

**Table 2 sensors-21-04443-t002:** Comparing results of Mocap-based style/subject recognition (HDM05 dataset). The average accuracy is reported concerning the cited works vs. our approach—EHECCO+SVM.

Method	Accuracy (%)
SPDNet [40]	61.45
SE [41]	70.26
SO [42]	71.31
LieNet [31]	75.78
Seq2Im+SVM [27]	70.70
Seq2Im+KNN [27]	66.82
Seq2IM+RF [27]	80.62
Seq2Im+CNN (fine-tuning) [27]	83.33
EHECCO+SVM	88.80

**Table 3 sensors-21-04443-t003:** Comparing results of Mocap-based action recognition (CMU subset database). The average accuracy is reported concerning the cited works vs. our approach—EHECCO+SVM.

Method	Accuracy (%)
MT+DTW [58]	82.9
SSM+DTW [18]	85.3
EMR [59]	86.7
MW+CNN [26]	90.7
EHECCO+SVM	90.0

## Data Availability

The databases used in this study are public and can be found at the following links: HDM05: http://resources.mpi-inf.mpg.de/HDM05/ (accessed on 5 October 2020), CMU subset: http://mocap.cs.cmu.edu/info.php (accessed on 5 October 2020), and Tennis-Mocap: https://github.com/jdpulgarin/Tennis-MoCap (accessed on 5 October 2020).

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
