# Peer review of "An Enhanced Joint Hilbert Embedding-Based Metric to Support Mocap Data Classification with Preserved Interpretability"

_sensors, 2021, doi:10.3390/s21134443_

Round 1

Reviewer 1 Report

This paper studies the problem about Mocap data classification. Specifically, this paper proposes an enhanced Hilbert embedding-based framework from a cross-covariance operator, termed (EHECCO) to represent and discriminate joint probability distributions in RKHS, which can be divided into two steps. First, a codebook generation and a PCA algorithm is performed to code the most relevant patterns from the Mocap series. Afterwards, EHECCO-based metric is computed and utilized in an SVM classifier. EHECCO framework shows strength in terms of the visualization and interpretability of the results.

The reviewer has some major concerns that need to be addressed in the next version of this paper as follows.

First, in Section 2.2, the authors perform a detailed deduction about the process of computing the EHECCO’s distance matrix. It would be more convincing if the authors could explain more about the motivation and physical meaning of the Enhanced Hilbert Embedding from cross-covariance operator. Specifically, why does EHECCO’s distance matrix more likely to perform well on time series classification issues.

Second, in Section 3.2, the authors consider three principal components while computing the latent space to code the most relevant orthonormal basis. It would be better if the authors provide more explanation about why they choose three as the number of principal components.

Third, in Section 3.2, the authors train a support vector machine (SVM) classifier on the EHECCO’s distance matrix. However, there are many nonlinear relationships and patterns in the data. Could the authors further explain why they choose a linear classifier? Would the performance become better on a nonlinear or more complex model?

Fourth, this paper fails to properly cite several past related works (e.g., [1-3]) and carefully compare the differences between them and this paper.

[1] Towards 3D Human Pose Construction Using WiFi, MobiCom 2020.

[2] SKEPRID: Pose and Illumination Change-Resistant Skeleton-Based Person Re-Identification, TOMM 2018.

[3] RFID-Pose: Vision-Aided Three-Dimensional Human Pose Estimation With Radio-Frequency Identification, IEEE Transactions on Reliability, 2020.

Fifth, in Section 4.2, the authors show t-SNE-based 2D projection from EHECCO distance for Tennis-Mocap database in Figure 6. However, based on the information revealed in Figure 6, it seems EHECCO distance do not distinguish smash from serve and ForeHand very well. Could the authors further provide some explanation about it?

Reviewer 2 Report

1) The paper proposes an enhanced joint Hilbert embedding-based metric to support MOCAP data classification that achieves competitive classification results for action recognition tasks.

The article has a good contribution, the main contribution is an enhanced Hilbert embedding-based approach from a cross-covariance operator EHECCO.

Linea 288-292. Good result, consider to use (Seq2im+CNN) instead of ", i.e., Seq2im+CNN":
"EHECCO+SVM combination overcomes the state-of-the art techniques compared, including those based on deep learning, i.e., Seq2im+CNN. Nevertheless, deep learning approaches often require exhaustive fine-tuning, whereas our EHECCO-based metric provides a data-driven technique as input vector evaluations for nonlinear pattern extraction in RHKS."

However, there are factors that prevent reading fluency and to appreciate the development and contribution:

2) On English grammar
From the Abstract section, some phrases like:

a) "Motion Capture-(Mocap) data are widely used time series to study human movement."
could be rewritten as:
"Motion Capture-(Mocap) data are widely used as time series to study human movement."

b) "Besides, the same human action may have variations because the individual alters the movement and the inter/intraclass variability." 
It seems that they take a literal translation from Spanish to English, an option could be:
"Besides, the same MOCAP for human action may have variations because the individual alters their movement and therefore the inter/intraclass variability."

c) Please consider changing
"... to map the input Mocap data to a tensor space built from both 3D skeletal joints and a principal component analysis based projection."
to
"... to map the input Mocap time series data to a tensor space built from both 3D skeletal joints and a principal component analysis based projection."

3) On reading fluency

When authors comment about "EHECCO favors the interpretation of relevant anthropometric variables correlated with player’s expertise"
it should be described in detail, an example is missing and it is appropriated.

Line 17. "Time series classification is a real-world problem that frequently deals with vast quantities of numerical measurements acquired at irregular time intervals"
refers to measurements acquired at irregular time, but time series are often studied or mapped to regular intervals.
This paragraph needs revision to express the idea in a better way.

Line 54. Authors say "DB approaches reside in the construction of a dissimilarity space from the input time series, which are later used to train a classifier, i.e., a K-nearest neighbors"
The paragraph requires revision. "Such as" or "for example" is better to use instead of "i.e." since i.e. means equivalent, but not all DB approaches are synonymous  of K-NN.

Line 57. ED is used but it was not yet defined, it seems that means Euclidean-based.
Line 58. The use of "l2 distances" requires a more extensive explanation.

Authors use a lot of acronyms, such as reproducing kernel Hilbert space for (RKHS) they should consider to capitalize the words, in this case Reproducing Kernel Hilbert Space.

Line 74. Authors state that Enhanced Hilbert embedding-based framework is proposed, but also authors refers to this kind of metrics as dis/similarities
so, the question is... what is a framework? is really EHEB a framework? (cf. see line 82, it is used dissimilarity)

Line 79. The paragraph requires revisions to provide reading fluency, since there is a leap from proposing a new metric-concept to directly justify a specific application. That is, there must be other applications that are being ommited, and that could be commented on, and then to clarify that it is also possible to use it for HAR.

Avoid using "the results" since "the" refers to all or specific results, I suggest "...of results concerning human action recognition (HAR)."

4) A description of what is the inspiration or motivation that yield authors to this EHECCO metric could help the reader to understand their approach (may be at the beginning of Section 2, and before section 2.1) now there is a "discontinuity" from an argumentative Section 1, to a formal section 2. The same from subsections 2.1 to 2.2.

5) Line 143. Is not trivial why and how " (5) and (8) favors the extraction of relevant patterns from joint distributions as vector based mappings in RKHS" at this point a Theorem or demonstrations is highly convenient to complement the schematic illustration of Figure 1. A similar comment is for the sentence of Lines 151 to 153.

6) Some spaces could be included to make clear the content of Table 1, beyond the colors. Capitalize all these terms.

7) An inclusion  of the general methodology that follow the paper is helpful. For example, space between Section 3 and subsection 3.1 is empty, focused on the databases, Mocao data classification, and so on, but they are technical  data, without a methodological context.

8) A Experiment design section could help to separate the technical details from data classification tasks, i.e. What to do and why? vs How to do it?
to complement the methodological approach the authors are following.

For example, Subsection 3.3 could be divided to show two experiments about method comparison for HDM05 dataset, and CMU subsets (to get a mind map).
Note that it could take to restructurate subsection 4.1 where HDM05 and CMU results are presented simultaneously (line 274-275).

9) Line 270. A citation for (t-SNE) algorithm is required.

10) Below Line 305. Table 2 Caption is too large. Some text should be written as paragraphs.

11) Remark (bold) the best result in Table 2, and the best of state of the art for comparison purposes.

12) Table 2 should be divided. One split for HDM05 and other for CMU.

Reviewer 3 Report

The content of the article is consistent with the scientific area of the journal Sensors. The subject raised by the authors is current and so far rarely noticed by other authors publishing in this area. The issue described may in the future contribute to improving the efficiency of the Hilbert embedding,  joint distribution, and time-series. Obtained results demonstrate how EHECCO represents and discriminates joint probability distributions as kernel-based evaluation of input time series. 
The paper has an original, scientific character, related to enhanced joint Hilbert embedding-based metric.
For a better clarification, please edit your paper as follows: 1. Extend the text of manuscript (example introduction or conclusion) to concrete results in the world and in Europe, - Improve the quality of the paper by presenting the results of publications of researchers and experts that are registered in the world databases (wos). They are specifically these:  Registration of Holographic Images Based on Integral Transformation and A new system for measuring the deflection of the beam with the support of digital holographic interferometry.  Thanks. 2. figure 1 should be contrasting and readable, 3. conclusions and future work should be extended to contain practical applications based on research described in this paper - expand references, 
4. highlight the course of dependencies/relations in figure No. 1 - 4; the purple color is indistinct,  5. Unify font in table No: 2, 
6. the paper should be read by a native english speaker. 
I recommend publishing the post after the proposed modifications.

Round 2

Reviewer 1 Report

The authors have addressed my comments to the previous version. I do not have further comments, and recommend that this paper be accepted.